# Element Content in Different Wheat Flours and Bread Varieties

**DOI:** 10.3390/foods11203176

**Published:** 2022-10-12

**Authors:** María Nerea Fernández-Canto, María Belén García-Gómez, Sonia Boado-Crego, María Lourdes Vázquez-Odériz, María Nieves Muñoz-Ferreiro, Matilde Lombardero-Fernández, Santiago Pereira-Lorenzo, María Ángeles Romero-Rodríguez

**Affiliations:** 1Areas of Nutrition and Food Science and Food Technology, Department of Analytical Chemistry and Food Science, Faculty of Science, University of Santiago de Compostela, 27002 Lugo, Spain; 2Modestya Research Group, Department of Statistics, Mathematical Analysis and Optimization, University of Santiago de Compostela, 27002 Lugo, Spain; 3Agronomy and Animal Science Group, Department of Anatomy, Animal Production and Veterinary Clinical Sciences, University of Santiago de Compostela, 27002 Lugo, Spain; 4Department of Plant Production and Engineering Projects, Escuela Politécnica Superior, Universidad de Santiago de Compostela, Galicia, 27002 Lugo, Spain

**Keywords:** ‘Caaveiro’, Galician bread, autochthonous cultivar, refined, wholegrain

## Abstract

The most consumed cereal-based product worldwide is bread. “Caaveiro”, an autochthonous variety with a recent growing interest, is one of the wheat varieties that fulfill the 25% local flour requirement in the PGI “Pan Galego” bread baking industry. The element content of the refined wheat flours used to make “Pan Galego” (‘‘Caaveiro’’, FCv; Castilla, FC; and a mixture of both, FM) was evaluated in ICP-MS. In addition, wholegrain flour (FWM) was included in the analysis. Loaves of bread were made with these flours (a, 100% FC; b, 100% FCv); and c, FM: 75% FC + 25% FCv) and their element content was analyzed. Wholegrain flour ranked the highest in almost all elements, highlighting the P (494.80 mg/100 g), while the FM and the FC presented the opposite behavior, with the highest Se values (14.4 and 15.8 mg/100 g, respectively). FCv was situated in an intermediate position regarding P, K, Mg, Mn, Zn, Fe and Na content, standing closer to FWM, although it presents the highest values for Cu (1076.3 µg/100 g). The differences observed in flour were maintained in bread. Hence, the local cultivar ‘‘Caaveiro’’ has an interesting nutritional profile from the point of view of the element content.

## 1. Introduction

Wheat (*Triticum aestivum*) is one of the most cultivated and consumed cereals in the world [1]. Cereal processing is an essential requirement to transform cereals into attractive and palatable bakery products. Those products are consumed throughout the world in their different varieties. Among them, bread is the most consumed. In fact, about 70–80% of wheat is consumed as bread [2], despite the progressive decrease in bread consumption observed in recent decades [3]. Wheat is the cereal for excellence in bread baking due to its superior baking performance when compared to other cereals [4,5]; moreover, wheat adapts to a wide range of growing conditions, while other cereals do not have this adaptability [6].

In addition to its indisputable sensory characteristics [7], bread is also interesting from a nutritional point of view. Regarding macronutrients, cereals that contain starch as the main polysaccharide, are an excellent source of carbohydrates (50–80% of the energy intake is carbohydrates). In the case of wheat, their protein content is 8–12 g/100 g, and supplies a high-quality provision of proline and glutamine, although it is poor in lysine. Lipids represent only 1.5–7.0 g/100 g of the cereal grain. As for micronutrients, especially wholegrain cereals, include mainly B vitamins, elements, antioxidants and phytosterols [4].

Furthermore, wheat grains derived from ancient wheat varieties are richer in resistant starch, fiber, elements, and phytochemicals than modern varieties. These nutritional characteristics make ancient wheat varieties interesting options in relation to the prevention of diseases such as high blood cholesterol, colitis and allergies [5], since these varieties have been associated with greater satiety as well as more favorable metabolic responses than breads made with modern and commercial wheat varieties [8]. The higher element content in breads made with ancient varieties has also been confirmed [9]. The autochthonous Galician (Northwest Spain) varieties such as the ‘Caaveiro’ variety can be placed within the ancient wheat varieties. ‘Caaveiro’ is one of the wheat varieties that fulfill the 25% (in relation to the total amount of flour) local flour requirement in the PGI “Pan Galego” bread baking industry.

Whole wheat grain is milled, and generally, only the endosperm is used to make a refined flour, since the bran and germ are discarded as by-products even though the whole grain is richer in protein, elements and vitamins, and the refined grain mainly contains starch [5]. Milling wheat into flour with a 70% extraction rate generally reduces element levels by between 60 and 80%. Furthermore, refined flour is mainly endosperm and contains 80–86% starch (dry basis), and has a very low content of dietary fiber and elements [6].

In contrast, the greatest amount of phytates is found in bran. Thus, breads made with wholemeal flour have a higher content of elements, but also phytates; consequently, the bioavailability of some elements is reduced [10]. Other studies suggest that the increase in the concentration of elements outweighs the negative effect induced by phytates on bioavailability [11].

Cereals contain about 1.5–2.5% elements, and P is the most abundant element in all cereals although is mostly affected by phytates. Wheat, rye, and oats are classified as rich sources of P (200–1200 mg per 100 g). K levels are also high in wheat, but cereal grains (before processing) are not considered to be a high dietary source of sodium. Wheat, rye, barley, and oats are also classified as moderate sources of Ca (100–200 mg/100 g), Mg (100–200 mg/100 g), Fe (1–5 mg per 100 g), Zn (1–5 mg/100 g) and Cu (0.1–1 mg/100 g). In addition to these elements, a large number of other elements are present in trace quantities.

Elements are essential micronutrients for human health [12] and wheat is notably an important source of these components, so the consumption of wheat flour and / or derived products plays an important role in sickness and health [13].

Sourdough, as a leavening agent, is the ancestor of modern yeast, although it is currently still used in the preparation of traditional breads such as PGI “Pan Galego”, originally from NW-Spain [14], and enriches the final product with a pleasant sour-taste [15] and a longer shelf life [16]. In addition to improving the sensory quality of bread [17], sourdough improves the bioavailability of elements [18] due to its ability to hydrolyze phytic acid [19] and levels of bioactive compounds [20]. It also reduces the glycemic index since it delays starch digestibility which leads to lower glycemic responses [21], and protects against health risks such as type 2 diabetes, obesity and cardiovascular disease, and other chronic dysmetabolic diseases [8]. Due to acidification, gluten degradation occurs and, therefore, decreases the inflammatory responses associated with celiac disease [18,22]. It even has potential as a prebiotic due to the extracellular polysaccharides produced by lactic acid bacteria [22]. In addition, it is a useful tool for masking and improving the acceptance of bread with a reduced salt content [20].

Since wheat bread is a widely consumed product and is part of the culinary culture in various regions, knowing its element contribution and how different wheat flours (because of the difference in variety and the milling process) influence the element content of bread is of interest to both the healthy population and also patients with certain pathologies. Thus, the aim of this work is to establish whether the element content of the flour is affected by flour variety or use of wholemeal flour, both in flour as a raw material, and in a manufactured product such as bread made with different leavening agents (sourdough and/or yeast) and fermentation times (2 and 12 h), usual conditions in Galician bakeries order to evaluate if these different conditions can cause differences in the content of mineral elements.

## 2. Materials and Methods

### 2.1. Raw Materials

The wheat flours used in the analysis and in the bread baking process were the Castilla-Spanish commercial variety (refined, ground in industrial mills) and the ‘Caaveiro’ variety (an autochthonous cultivar of Galicia-NW Spain and ground in stone mills); they have used alone and mixed together. Therefore, the wheat flour samples used were: (a) 100% Castilla variety (FC), (b) 100% ‘Caaveiro’ variety (FCv), (c) mixed flour (75% Castilla + 25% ‘Caaveiro’) (FM). Finally, a sample of wholegrain flour was also included: (d) homologous mixed wholegrain flour (75% Castilla + 25% ‘Caaveiro’) (FWM). Flour samples were provided by the DaCunha group.

Yeast, sourdough, and a mixture of both were used as leavening agents. Two sourdough types were used: sourdough, made from refined wheat flour (pH = 4.0) and sourdough made from wholegrain flour (pH = 4.2). The latter was used to make wholegrain breads.

Common commercial salt was used.

### 2.2. Bread Baking Procedure

The breads were made with the 4 types of flour described above, water, yeast (yeast breads), sourdough (sourdough breads) or sourdough and yeast (mixed leavening breads), and salt. One loaf of each type of bread has been made.

The dough formulation for yeast breads was flour (280.0 g), water (210.0 g), yeast (5.0 g) and salt (5.0 g). For sourdough breads, it was flour (225.7 g), water (169.3 g), sourdough (99.8 g) and salt (5.2 g). For the mixed leavening batches, the dough formulation included flour (239.5 g), water (179.6 g), sourdough (74.7 g), yeast (1.2 g) and commercial NaCl Hacendado (5.0 g). The total amount of dough was 500 g. For each type of bread, the total amount of dough was 500 g.

The sourdough was made mixing the corresponding flour, hydrated to 75% and leaving it at 25 °C for 3 days. From the 3rd day, it was renewed every day, taking half the weight, and completing the same weight with new water and flour. We waited for it to double in volume and kept it refrigerated until the following day.

The ingredients were mixed and kneaded for 30 min using the kneading program of a domestic bakery (Bifinett^®^ KH 2232). The short fermentation (2 h) was carried out at a controlled temperature of 23 °C ± 2 and humidity of 45 ± 5%. The long one (12 h) was carried out at 12 °C ± 2. Loaves were manually round-shaped and baked at 180 °C for 30 min. in a domestic oven (Carrefour^®^).

The resulting breads are shown in Table 1.

### 2.3. Determination of Elements Content

#### 2.3.1. Sample Preparation

The wheat flour and bread (crust and crumb) samples were dried at 130 ± 3 °C according to the official AOAC method 925.10 [23]. Each sample was dried in duplicate. From the dry samples, around 0.800 g of each sample were weighed, and 3 mL of 69% HNO_3_ (Hiperpur, Panreac^®^) and 2 mL of deionized H_2_O (Milli-Q) were added. Microwave digestion (Milestone, Ultrawave) was performed at 240 °C, and 40 bars for 40 min at 1500 W. Once digested, they were filled to a final volume of 50 mL with H_2_O (Milli-Q). The macro- (Ca, Mg, Na, K and P) and microelements (Cu, Fe, Mn, Zn and Se) were determined.

#### 2.3.2. Analysis of Elements in ICP-MS

The equipment used was an ICP-MS (Agilent 7700×) with a sample introduction system consisting of a Micromist glass low-flow nebulizer, a double-pass glass spray chamber with a Peltier system (2 °C) and a quartz torch. Measurements were taken five times.

The calibration standards were prepared in HNO_3_-H_2_O, in the same proportion as in the samples. As calibration standards, the multi-elemental standard Multi IV (Merck^®^) in concentrations between 0.2–10,000 µg·L^−1^, P and Ca standard (Panreac^®^) in concentrations between 0.1–10 mg·L^−1^ and Selenium (Merck^®^) in concentrations between 0.2–100 µg·L^−1^ were used. The correlation coefficients of the calibration lines for each element were equal to or greater than 0.999.

#### 2.3.3. Statistical Analysis

A multivariate approach using Principal Component Analysis (PCA) was applied in order to carry out a graphical representation in low dimension space. The distances between samples of flour or breads were analyzed with 90% confidence ellipses on the first two dimensions resulting from a PCA of the element content dataset.

Computing the mean on the element content dataset is not a good location measure to characterize the samples because of the influence of outliers or highly skewed distributional shapes. In this case, one strategy is to use more robust measures such as the trimmed mean and perform tests based on the corresponding sampling distribution of such robust measures. The 20% trimmed means and standard error were computed and used in robust variants of one-way ANOVA. An appealing feature of a 20% trimmed mean is that it achieves nearly the same amount of power as the mean when sampling from a normal distribution, and when there are outliers, a 20% trimmed mean can present a substantially smaller standard error [24].

Hierarchical clustering was carried out to classify samples of breads according to the element content. Hierarchical clustering is a cluster analysis method which aims at identifying a series of clusters within a nested structure. It produces a tree-based representation (i.e., dendrogram) of sorted data. Objects in the dendrogram are linked together based on their similarity. The Euclidean distance of scaled data was used to calculate the degree of similarity between each pair of samples and Ward’s minimum variance method was the agglomeration (linkage) method used to compute the distance between clusters. It minimizes the total within-cluster variance. In the dendrogram displayed. The height of the fusion into branches, provided on the vertical axis, indicates the (dis)similarity between the two clusters. The higher the height of the fusion, the less similar the clusters are. This height is known as the cophenetic distance between the two clusters, one way to measure how well the cluster tree generated reflects the original data is to compute the correlation between the cophenetic distances and the original distance data.

The description of the clusters was made using a v-test (Test Values) calculated for each element so that the list of significant elements was provided for each cluster. Test Values are measurements of the distance between the within-class value and the overall value [25]. The cluster information was added onto the factorial plane in PCA by using a color code.

The R package ecosystem [26] was used to apply robust methods and multivariate analysis by means of WRS2, FactoMineR and Factoextra packages.

## 3. Results and Discussion

### 3.1. Flour Element Content

The element content of the flours analyzed is shown in Table 2.

Principal Components Analysis was performed with the data obtained after analyzing the 4 flours used in this study in order to provide a data structure in a reduced dimension, covering the maximum amount of information present in the data (Figure 1). The first two principal components accounted for 95.5% of the total variance. The cumulative variance was regarded as enough to distinguish the flours by their element content. Figure 1 shows that all elements were dominant variables in the first principal component (PC1) that accounted for 83.7% of the total variance, although Cu showed the lowest values. The dominant variables appeared at positive values of PC1, except Se. Principal component two (PC2) explained up to 11.8% of the total variance. Examining the loadings for this PC, Cu appeared as the most dominant variable on the negative side of PC2.

When examining the score plot of the samples in the space defined by the two principal components considered (Figure 1), a separation between refined-wholegrain flours was observed due to their contrasting behavior in terms of the element content. With the exception of Se, FWM showed higher element content, and the refined flours (FM and FC) presented the lowest element content except Se. For Se, the behavior was the opposite; Se was higher in the case of FM and FC flours and lowered in the case of FWM.

FCv was situated in an intermediate position between FWM and the refined flours (FC and FM) in dimension 1 and located in the negative part of dimension 2 due to the strong influence of Cu, and to a lesser extent Ca, since FCv was the flour with the highest Cu content and the lowest Ca content.

In order to compare the means, a robust variant of one-way ANOVA based on trimmed means (20% trimming level) was applied (Table 2).

The most abundant element in the flours tested was P, followed by K. Regarding microelements, Mn, Zn and Fe were clearly the ones observed in higher proportions. These results concurred with those obtained in previous studies in both refined and wholegrain wheat flours [13,27].

When the literature was reviewed, a great diversity of results was found. The differences observed when comparing the results of the present study with the results reported by the different researchers might be due to the differences in the wheat varieties analyzed, the edaphoclimatic variables associated with different locations, harvest years, etc. [11,13,28,29,30,31].

Thus, regarding the content of element in wholegrain flours, element values lower than those obtained for FWM were found by different researchers [27,32], ([27]: P (235.25 mg/100 g), Ca (36.18 mg/100 g), Mg (118.47 mg/100 g), Fe (4.073 mg/100 g), Cu (290 µg/100 g), Zn (3.22 mg/100 g), Mn (2.29 mg/100 g) and Se (7.6 µg/100 g) and [32]: Ca (31 mg/100 g), all on d.w.), except for K (422.91 mg/100 g d.w.) and Na (10.64 mg/100 g d.w.) [27]. A study reported similar or lower element values for refined flours (Fe: 1.30 mg/100 g, Mn: 0.74 mg/100 g, Zn: 1.39 mg/100 g and Cu: 300 µg/100 g d.w.) [33] compared with those observed in the present study for FC, FM and FCv.

The results of a post-hoc Yuen’s test when comparing the 4 flour samples were statistically significant (*p*-value < 0.05) for all elements. According to the results, wholegrain mixed flour (FWM) showed a significantly higher content for all elements, except Cu and Se. All in all, the array of the high element content in flours followed this order, FWM was first, followed by FCv and then, FM and FC with statistically significant differences as shown in Table 2. The only exception appears for Ca, in which FCv moved from second to fourth place.

The values obtained for P, K, Mg, Mn, and Zn in FWM samples were essentially three times more than those obtained in the FC and FM ones. In addition, the values in the FWM samples were almost double those obtained in FCv (Table 2). These results were as to be expected since the ‘Caaveiro’ flour was ground in stone mills, and, thus, retained part of the bran. The differences demonstrated between refined and wholegrain flours in this study coincided with the ones obtained by different authors as described below.

Previous studies that compared a wholegrain with two refined flours reported P and K values that were between two and three times higher for wholegrain flour than for refined flour (P: 390 vs. 101 and 130 mg/100 g, K: 360 vs. 101 and 130 mg/100 g d.w.) [31]; in some cases, the K values reported were even three/four times as high (553.5 vs. 125.1 mg/100 g and 279.66 vs. mg/100 g d.w.) [13,30].

Following the same trend as that observed in the present study, some previous investigations revealed pronounced differences between refined and whole wheat flour regarding other elements. Thus, Mn values showed six times as high in whole wheat as compared to refined flour [13,31] (0.3 vs. 2.4 mg/100 g and 0.43/0.73 vs. 3.7 mg/100 g d.w., respectively), Mg values were four times as high (21/40 vs. 140 mg/100 g and 25.95 vs. 108.64 mg/100 g d.w., respectively) [30,31] and Zn values four times as high were also found (0.4–0.5 vs. 1.7–2.0 mg/100 g d.w.) [29]. In other study, even though a higher average content of Zn was displayed in whole grain flours when compared to refined ones, the differences were smaller (Zn: 1.08 vs. 1.86 mg/100 g d.w.) [30]. Nevertheless, when analyzing different varieties of Pakistani spring wheat, [28] obtained a lower proportion of Zn in whole wheat flour than in refined flour in two cultivars (2.6 vs. 3.20 and 2.1 mg/100 g vs. 3.35 mg/100 g d.w.).

Regarding the Ca (Table 2), the highest and lowest values have been observed in FWM and in FCv, respectively (the FWM value almost doubled the FCv one). FC and FM manifest the same values which were significantly different from FWM and FCv. Vignola et al. [30] reported Ca values in wholegrain flour that were double those obtained in refined flour (14.60 vs. 32.41 mg/100 g d.w.). The difference in Ca content between wholemeal (22.1 mg/100 g d.w.) and refined flour (15.2 mg/100 g d.w.) was not as remarkable [13]. Nonetheless others reported a significantly lower Ca content in whole wheat flour (0.31 mg/100 g d.w.) than in refined flour (0.66 mg/100 g d.w.) [32].

Fe values were significantly higher in FWM that in FC and FM. However, in FCv intermediate values were obtained that did not noticeably differ from the other flours analyzed. Likewise, in previous investigations large differences were observed in Fe values in wholegrain as compared to refined flours [13,29,30,31] ([13]: 4.4 vs. 0.7 mg/100 g, [29]: 3.0–4.8 vs. 0.6–1.2 mg/100 g, [30]: 2.46/2.10 vs. 0.71/1.2 mg/100 g and [31]: 3.7 vs. 0.71/1.1 mg/100 g, all on d.w.).

The Na content was patently higher in the FWM than in the rest of the flours analyzed. The Na content reported in other investigations was significantly higher in whole wheat flour than in refined flour and, much like the present study, the Na values in the whole wheat flour were three times those obtained in the refined flour; however, the Na values reported by these researchers were twice as high as the values observed for both refined and wholegrain flour in the study at hand (2 vs. 6.5 mg/100, d.w.) [32]. In addition, another study observed higher Na contents in whole wheat flour (0.66 mg/100 g d.w.) than in two refined flours (0.34/0.55 mg/100 g d.w.) [31]. In sharp contrast, for a certain variety and for a specific harvest year [13], the Na value was lower in whole wheat flour (24.3 mg/100 g d.w.) than in refined flour (43.6 mg/100 g d.w.).

In the case of Cu, a significantly higher value has been observed in FCv than in the rest of the flours; this is true even in FWM, the wholegrain flour. In the same way, different studies showed higher Cu values in wholegrain flours than in refined ones; in these cases, the degree of difference found depended on the variety and harvest: 161/295 vs. 215/320 µg/100 g [30] and 140/180 vs. 440 µg/100 g d.w. [31].

Finally, in the observation of the remaining elements analyzed, a clear distinction was found for Se, which had a noticeably higher proportion in FC and FM than in FCv and FWM. However, in this case, there was no marked difference between the FC and FM or between FCv and FWM, respectively. In contrast to these results, other researchers reported higher Se contents in a whole wheat flour (3.9 µg/100 g d.w.) than in two refined flours (0.74/1.3, d.w.) [31].

### 3.2. Bread Element Content

The results obtained for the breads analyzed are shown in Table 3.

Cluster analysis was used as an exploratory analysis method to determine element content. Breads were clustered to suggest connections between samples. To assess the clustering tendency of the dataset, the Hopkins statistic (H) was used. It is based on measuring the probability that the dataset of element content in breads was generated by a uniform data distribution (i.e., no meaningful clusters). This dataset formed by 10 variables, one for each metal analyzed, is highly clusterable. The H value = 0.18 and was far below the H threshold of 0.5.

In the agglomerative hierarchical cluster (Figure 2) each bread was initially considered as a single-element cluster (leaf). At each step of the algorithm, the two most similar clusters were combined into a new bigger cluster (nodes). This procedure was iterated until all points were a member of just one single big cluster (root). The result was a tree which can be displayed using a dendrogram (Figure 2).

The correlation between the cophenetic distances and the original distance data was computed to measure how well the cluster tree generated reflected the original data. When the clustering is valid, the linking of objects in the cluster tree should have a strong correlation with the distances between objects in the original distance matrix. The closer the value of the correlation coefficient is to 1, the more accurately the clustering solution reflects the data. Values above 0.75 are considered to be good. For the element dataset of the present study, the value of the correlation coefficient was 0.89. These results show that the variability in terms of the total element content was due to the type and origin of the flour used in the bread preparation rather than to the leavening agent or the fermentation time.

The v-test was used to describe each group of breads. In Table 3, the elements were sorted according to two keys: the sign of the difference between the average score for each group of breads and the overall mean, and the significance of the *p*-value from the test that compared the mean with the samples of the cluster to the overall mean. The elements were organized using the descending order provided by the v-test by only considering *p*-values less than 5% to be significant.

Cluster 1 was composed of wholegrain flour breads (BWM2S, BWM12S, BWM2M, BWM12M, BWM2Y and BWM12Y), which displayed a high value in most of the elements except Se. The high values found here were somewhat higher than the global average for each element, as reflected by the positive sign of the v-test (Table 2). Moreover, these breads had the lowest Se level. The *p*-values provide an indication of the “significance” of a given deviation of the general average and in this cluster and in this group the *p*-values associated with all elements were particularly small compared to the one associated with Cu.

A research where two wholegrain breads were analyzed, one made with a spontaneously fermenting sourdough and the other made with commercial yeast, and obtained results that showed mean Zn (sourdough: 0.68 mg/100 g and yeast: 1.14 mg/100 g d.w.), Mg (sourdough: 28.88 mg/100 g and yeast: 24.78 mg/100 g d.w.), K (sourdough: 85.87 mg/100 g and yeast: 84.78 mg/100 g d.w.) and Ca (sourdough: 4.11 mg/100 g and yeast: 8.06 mg/100 g d.w.) values lower than BWM breads [34].

In addition, others examined the range values for 12 wholegrain and sourdough breads [35] and observed ranges of Ca (37.81–43.47 mg/100 g), Fe (4.53–5.84 mg/100 g), Zn (1.68–2.48 mg/100 g) and Mg (84.31–93.13 mg/100 g). These values are higher than in the previous study [34], but they are still lower than those obtained in the present study for BWM breads since the authors express the results in dry matter.

Cluster 2 is composed of ‘Caaveiro’ flour breads that can be described as a sort of average bread as regards all elements, except Cu, Ca, and Se. ‘Caaveiro’ flour breads show higher than average values for Cu and below average values for Ca and Se. As previously observed for the different types of flour, bread made from ‘Caaveiro’ flour showed characteristics that appear between those found for whole wheat flour and refined flours because the flour that is used to make PGI bread “Pan Galego” is traditionally ground in stone mills and retains part of the bran.

In cluster 3, composed of breads made with refined flours (BM2S, BM12S, BM2M, BM12M, BM2Y, BM12Y, BC2S, BC12S, BC2M, BC12M, BC2Y, BC12Y), the behavior of the elements was opposite that of the wholegrain breads. This category was characterized by breads having a below-average element content as evidenced in the negative v-test for all of them except Se. In these breads, Ca presented the least significant *p*-value.

In the element contents obtained in 16 pre-baked breads made with refined flour using a mixture of sourdough and yeast or only yeast as a leavening agent [36] was observed that P, K and Mg were the most abundant elements in the breads. The average values on d.w. for the P (151.2 mg/100 g), Mg (35.4 mg/100 g), Ca (26.6 mg/100 g), Fe (1.47 mg/100 g), Mn (1.31 mg/100 g), Zn (1.13 mg/100 g), K (196.6 mg/100 g) and Cu (352.0 µg/100 g) observed in these 16 pre-baked breads were lower than those obtained in the refined breads (BM and BC).

In addition, when comparing the results obtained in the present study with those obtained in previous research, who also determined elements in Portuguese refined wheat breads [37], the values of K, P, Mg, Fe, Mn, Zn and Cu obtained in BM and BC breads were within the range reported by these researchers.

To visualize the behavior of the different bread samples in terms of element content, a PCA analysis was performed (Figure 3). The variance explained by the first two main components accounted for 88.5%, where 67.2% corresponded to PC1. The position in the plane illustrated the characterization that was rendered in the v-test table (Table 2). The FWM breads were situated in the positive part of dimension 1, and were characterized, as mentioned above, by high values in all elements except Se. The breads made from the refined flours (FC and FM breads) were located on the opposite side and FCv breads were located in an intermediate position between the FC and FM and FWM breads. In this way then, it is patent that the PCA provided an element distribution in breads that was very similar to the one obtained in flours (Figure 1), with the exception of Na, as expected, since NaCl was added during bread baking process.

In a previous study, was observed that bread made with wholegrain flour contained more Ca (36.0 vs. 88.0 mg/100 g d.w.), Mg (102.0 vs. 32.0 mg/100 g d.w.), Fe (4.4 vs. 1.7 mg/100 g d.w.), Cu (400 vs. 200 µg/100 g d.w.) and Zn (2.7 vs. 0.11 mg/100 g d.w.) than bread made with refined flour [38].

Other investigation also reported a higher content of Mg (48 vs. 13 mg/100 g d.w.), K (386 vs. 205 mg/100 g d.w.), Ca (75 vs. 19 mg/100 g d.w.), Mn (1.26 vs. 0.62 mg/100 g d.w.), Fe (2.25 vs. 1.05 mg/100 g d.w.) and Zn (1.53 vs. 0.78 mg/100 g d.w.) in wholegrain bread than in refined bread [39]. These researchers showed a higher Cu content in wholegrain bread than in refined one (390 vs. 220 µg/100 g d.w.), a trend that was true for BC and BM breads, but not for BCv breads, which presented Cu values higher than those in BWM breads. The values obtained for BCv and BWM breads were higher than those reported for the wholegrain bread analyzed by these researchers.

The study performed by [40] confirmed that wholegrain bread contained more K and Mg than refined bread (K: 212.3 vs. 201.4 mg/100 g and Mg: 48.0 vs. 26.7 mg/100 g). They reported K and Mg values lower than those obtained in all the breads analyzed in the present study. Contrary to the trend observed in the present study and in previous studies, they found a lower Ca content in bread made with wholegrain flour than in bread made with refined flour (19.6 vs. 62.0 mg/100 g).

## 4. Conclusions

The type and origin of the flour had an important influence on the total element content, both in the flour itself and in the breads.

Wholegrain flour (FWM) showed the highest value for 7 of the 10 elements analyzed (Mg, P, K, Ca, Na, Mn, Fe and Zn) and the lowest value for Se.

The refined flours (FC and FM) have the opposite behavior, they showed the lowest values for the 7 elements mentioned and the highest value for Se.

The Galician local cultivar flour (FCv) used in this study stands out as having the lowest Ca and the highest Cu content. The element content of Mg, P, K, Ca, Na, Mn, Fe and Zn in FCv was found to be between the values displayed for wholegrain (FWM) and refined (FC and FM) flours; however, the FCv values were closer to those found for FWM.

In the analysis of the presence of these elements in breads made with the different flours and methods, the same behavior observed for flours was observed in the bread itself, with the exception of the appearance of Na, which happened because NaCl was added during the bread making process. This finding endorses the inclusion of the ‘Caaveiro’ flour variety, an autochthonous Galician cultivar (Northwest Spain) in the PGI “Pan Galego” bread baking industry since it provides strong evidence of its nutritional quality.

## Figures and Tables

**Figure 1 foods-11-03176-f001:**
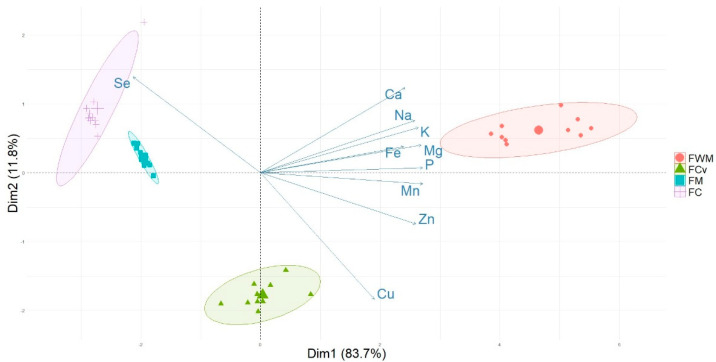
Principal components analysis (PC1 vs. PC2) describing the element variation between the flours and 90% confidence ellipses. FC: 100% refined Castilla variety; FCv: 100% refined ‘Caaveiro’ variety; FM: 75% FC + 25% FCv refined mixed flour: FWM: 75% FC + 25% FCv wholemeal mixed flour.

**Figure 2 foods-11-03176-f002:**
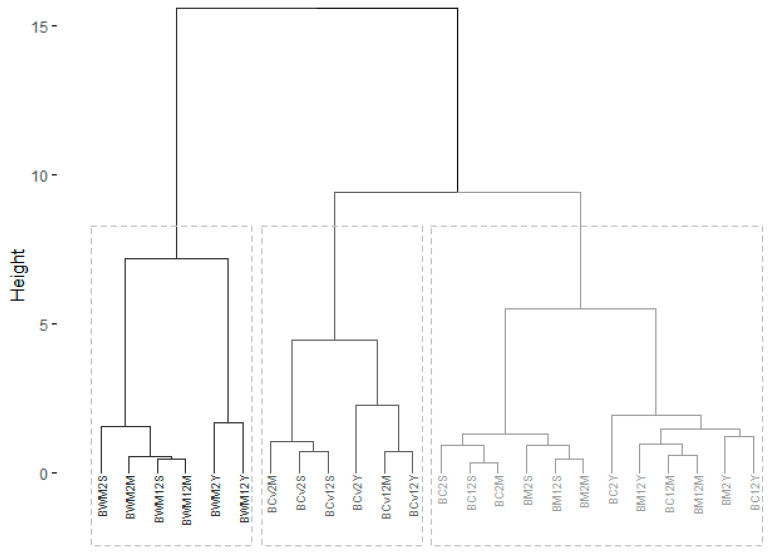
Dendogram for cluster analysis (BWM2S: 75% FC + 25% FCv wholemeal mixed flour/2 h fermentation/sourdough; BWM2M: 75% FC + 25% FCv wholemeal mixed flour/2 h fermentation/mixed leavening; BWM2Y: 75% FC + 25% FCv wholemeal mixed flour/2 h fermentation/yeast; BWM12S: 75% FC + 25% FCv wholemeal mixed flour/12 h fermentation/sourdough; BWM12M: 75% FC + 25% FCv wholemeal mixed flour/12 h fermentation/mixed leavening; BWM12Y: 75% FC + 25% FCv wholemeal mixed flour/12 h fermentation/yeast; BM2S: 75% FC + 25% FCv refined mixed flour/2 h fermentation/sourdough; BM2M: 75% FC + 25% FCv refined mixed flour/2 h fermentation/mixed leavening; BM2Y: 75% FC + 25% FCv refined mixed flour/2 h fermentation/yeast; BM12S: 75% FC + 25% FCv refined mixed flour/12 h fermentation/sourdough; BM12M: 75% FC + 25% FCv refined mixed flour/12 h fermentation/mixed leavening; BM12Y: 75% FC + 25% FCv refined mixed flour/12 h fermentation/yeast; BCv2S: 100% refined ‘Caaveiro’ variety/2 h fermentation/sourdough; BCv2M: 100% refined ‘Caaveiro’ variety/2 h fermentation/mixed leavening; BCv2Y: 100% refined ‘Caaveiro’ variety/2 h fermentation/yeast; BCv12S: 100% refined ‘Caaveiro’ variety/12 h fermentation/sourdough; BCv12M: 100% refined ‘Caaveiro’ variety/12 h fermentation/mixed leavening; BCv12Y: 100% refined ‘Caaveiro’ variety/12 h fermentation/yeast; BC2S: 100% refined ‘Castilla’ variety/2 h fermentation/sourdough; BC2M: 100% refined ‘Castilla’ variety/2 h fermentation/mixed leavening; BC2Y: 100% refined ‘Castilla’ variety/2 h fermentation/yeast; BC12S: 100% refined ‘Castilla’ variety/12 h fermentation/sourdough; BC12M: 100% refined ‘Castilla’ variety/12 h fermentation/mixed leavening; BC12Y: 100% refined ‘Castilla’ variety/12 h fermentation/yeast).

**Figure 3 foods-11-03176-f003:**
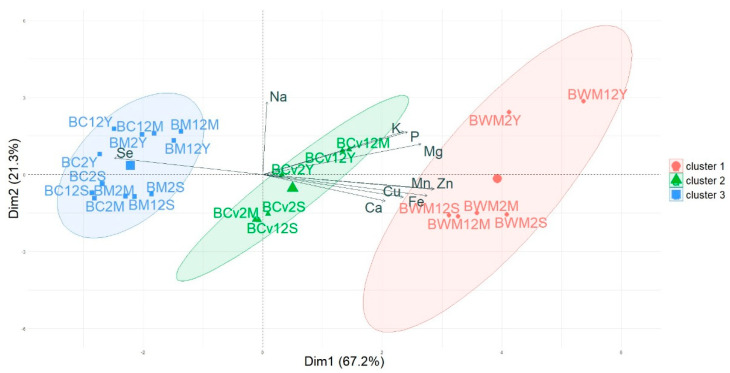
Principal components analysis (PC1 vs. PC2) describing the variation among the breads manufactured with different wheat flours (Group 1: BWM breads, group 2: BCv breads and group 3: BC and BM breads). BWM2S: 75% FC + 25% FCv wholemeal mixed flour/2 h fermentation/sourdough; BWM2M: 75% FC + 25% FCv wholemeal mixed flour/2 h fermentation/mixed leavening; BWM2Y: 75% FC + 25% FCv wholemeal mixed flour/2 h fermentation/yeast; BWM12S: 75% FC + 25% FCv wholemeal mixed flour/12 h fermentation/sourdough; BWM12M: 75% FC + 25% FCv wholemeal mixed flour/12 h fermentation/mixed leavening; BWM12Y: 75% FC + 25% FCv wholemeal mixed flour/12 h fermentation/yeast; BM2S: 75% FC + 25% FCv refined mixed flour/2 h fermentation/sourdough; BM2M: 75% FC + 25% FCv refined mixed flour/2 h fermentation/mixed leavening; BM2Y: 75% FC + 25% FCv refined mixed flour/2 h fermentation/yeast; BM12S: 75% FC + 25% FCv refined mixed flour/12 h fermentation/sourdough; BM12M: 75% FC + 25% FCv refined mixed flour/12 h fermentation/mixed leavening; BM12Y: 75% FC + 25% FCv refined mixed flour/12 h fermentation/yeast; BCv2S: 100% refined ‘Caaveiro’ variety/2 h fermentation/sourdough; BCv2M: 100% refined ‘Caaveiro’ variety/2 h fermentation/mixed leavening; BCv2Y: 100% refined ‘Caaveiro’ variety/2 h fermentation/yeast; BCv12S: 100% refined ‘Caaveiro’ variety/12 h fermentation/sourdough; BCv12M: 100% refined ‘Caaveiro’ variety/12 h fermentation/mixed leavening; BCv12Y: 100% refined ‘Caaveiro’ variety/12 h fermentation/yeast; BC2S: 100% refined ‘Castilla’ variety/2 h fermentation/sourdough; BC2M: 100% refined ‘Castilla’ variety/2 h fermentation/mixed leavening; BC2Y: 100% refined ‘Castilla’ variety/2 h fermentation/yeast; BC12S: 100% refined ‘Castilla’ variety/12 h fermentation/sourdough; BC12M: 100% refined ‘Castilla’ variety/12 h fermentation/mixed leavening; BC12Y: 100% refined ‘Castilla’ variety/12 h fermentation/yeast.

**Table 1 foods-11-03176-t001:** Flour composition and characteristics of resulting breads.

Flour	Bread *			
	Fermentation Time (h)	Leavening Agent		
		Yeast	Sourdough	Mixed Leavening
100% refined Castilla variety (FC)	2	BC2Y	BC2S	BC2M
	12	BC12Y	BC12S	BC12M
100% refined ‘Caaveiro’ variety (FCv)	2	BCv2Y	BCv2S	BCv2M
	12	BCv12Y	BCv12S	BCv12M
75% FC + 25% FCv refined mixed flour (FM)	2	BM2Y	BM2S	BM2M
	12	BM12Y	BWM12S	BWM12M
75% FC + 25% FCv wholemeal mixed flour (FWM)	2	BWM2Y	BWM2S	BWM2M
	12	BWM12Y	BWM12S	BWM12M

* The first letter (B) is for bread; the next is the type of flour used for its preparation (Castilla, C; ‘Caaveiro’, Cv; refined mixed, M; or wholemeal mixed, WM); the number is the fermentation hours (2 or 12); and the last letter is the leavening agent (yeast, Y; sourdough, S; or mixture, M).

**Table 2 foods-11-03176-t002:** Element content (trimmed mean and standard error) of ten replicates for elements in the wheat flours on a dry weight basis (d.w.). Moisture means were: 10.9% (FWM), 10.0% (FCv), 9.6% (FM) and 10.3% (FC).

	FWM (Mixed Wholegrain Flour) Mean (Standard Error)	FCv (‘Caaveiro’ Flour) Mean (Standard Error)	FM (Mixed Refined Flour) Mean (Standard Error)	FC (Castilla Flour) Mean (Standard Error)
P (mg/100 g)	494.8 ^a^ (16.3)	302.2 ^b^ (5.2)	209.0 ^c^ (0.4)	174.1 ^d^ (0.7)
K (mg/100 g)	419.2 ^a^ (11.4)	191.4 ^b^ (3.0)	159.3 ^c^ (0.4)	146.8 ^d^ (0.8)
Mg (mg/100 g)	123.1 ^a^ (4.4)	61.5 ^b^ (0.9)	46.2 ^c^ (0.1)	39.6 ^d^ (0.3)
Ca (mg/100 g)	73.0 ^a^ (2.7)	36.0 ^c^ (0.5)	38.1 ^b^ (0.2)	38.4 ^b^ (0.2)
Mn (mg/100 g)	9.2 ^a^ (0.4)	4.9 ^b^ (0.1)	2.3 ^c^ (0.0)	1.4 ^d^ (0.0)
Zn (mg/100 g)	6.9 ^a^ (0.3)	5.2 ^b^ (0.1)	2.7 ^c^ (0.0)	1.3 ^d^ (0.0)
Fe (mg/100 g)	6.2 ^a^ (0.1)	3.1 ^abc^ (0.9)	3.0 ^b^ (0.0)	2.1 ^c^ (0.0)
Na (mg/100 g)	3.4 ^a^ (0.0)	1.5 ^b^ (0.1)	1.3 ^c^ (0.0)	1.3 ^b^ (0.0)
Cu (µg/100 g)	880.8 ^b^ (27.5)	1076.3 ^a^ (5.0)	410.9 ^c^ (0.9)	254.9 ^d^ (2.0)
Se (µg/100 g)	9.1 ^b^ (0.3)	10.3 ^b^ (0.4)	14.4 ^a^ (0.5)	15.8 ^a^ (0.3)

Different letters indicate statistically significant differences according to a post-hoc Yuen’s test.

**Table 3 foods-11-03176-t003:** Results for the v-test. Significant elements for characterization of bread clusters. Moisture means were: 41.2% (BWM breads), 40.39% (BCv breads), 37.5% (BM and BC breads).

Element	Mean in Category	SD in Category	v−Test	*p*−Value
Cluster 1. Wholegrain breads (BWM breads)				
Mn (mg/100 g)	4.8	0.3	4.4	0.0000
Ca (mg/100 g)	41.7	5.7	4.3	0.0000
Zn (mg/100 g)	3.6	0.2	4.0	0.0001
Mg (mg/100 g)	79.7	19.0	4.0	0.0001
Fe (mg/100 g)	3.9	0.5	3.9	0.0001
K (mg/100 g)	297.6	99.3	3.5	0.0004
P (mg/100 g)	345.6	106.1	3.2	0.0015
Cu (µg/100 g)	484.8	36.0	2.3	0.0221
Se (µg/100 g)	5.0	0.3	−3.0	0.0025
Cluster 2. ‘Caaveiro’ breads (BCv breads)				
Cu (µg/100 g)	525.5	49.1	3.0	0.0026
Ca (mg/100 g)	22.7	2.6	−2.0	0.0444
Se (µg/100 g)	5.6	0.7	−2.1	0.0321
Cluster 3. Refined breads (BM and BC breads)				
Se (µg/100 g)	8.8	0.7	4.5	0.0000
Ca (mg/100 g)	25.3	2.4	−2.0	0.0478
K (mg/100 g)	134.7	41.3	−2.4	0.0184
P (mg/100 g)	169.9	47.3	−2.8	0.0053
Mg (mg/100 g)	34.0	7.7	−2.3	0.0032
Mn (mg/100 g)	1.1	0.2	−3.9	0.0001
Fe (mg/100 g)	1.6	0.3	−4.0	0.0001
Zn (mg/100 g)	1.0	0.2	−4.0	0.0000
Cu (µg/100 g)	208.5	41.4	−4.6	0.0000

## Data Availability

The data used to support the findings of this study can be made available by the corresponding author upon request.

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
