# Peer review of "Element Content in Different Wheat Flours and Bread Varieties"

_foods, 2022, doi:10.3390/foods11203176_

Round 1

Reviewer 1 Report

Reviewer comments

I am grateful for the opportunity to review this piece of work. In this work, Fernández-Canto et al. determined the mineral composition of ancient wheat cultivars from Castilla variety (refined, ground in industrial mills) and the Caaveiro variety (ground in stone mills) used either alone and mixed together and their corresponding bread formulated from these wheat flours. The findings of the study are important as hidden hunger (micronutrient deficiencies) is common public health challenge. Bread is a commonly consumed staple food across the globe. Consequently, it could be targeted as a promising medium for the delivery of micronutrients to address nutritional challenges associated with micronutrients despite its rich caloric contribution from its carbohydrate provision. Please, find below my comments for your perusal.

Abstract

The wheat flour and bread formulations should be expanded and highlighted such as presented in the method “(a) 100% Castilla variety (FC), b) 100% Caaveiro variety (FCv), c) mixed flour (FM) (75% Castilla + 25% Caaveiro)”. The method for the mineral analysis should also be stated. For the results, the authors should state the concentration of predominant minerals that were identified in the flour from the wheat varieties and their corresponding bread.

Introduction

Line 29: The authors should introduce “of” before “the”

Line 43: The authors should revise this “As for micronutrients, cereals include mainly B vitamins,…………….”. It is wholegrain cereals that are rich in micronutrients, antioxidants and phytosterols from the bran oil.

Line 45: Revise “Furthermore, the wheat grains deriving from ancient wheat varieties….” to “Furthermore, wheat grains derived from ancient wheat varieties…………”

Line 45-46: Please, provide a reference for this statement “Furthermore, the wheat grains deriving from ancient wheat varieties are richer in resistant starch, fiber, minerals, and phytochemicals than modern varieties.”

Line 76-77: Is it the hole wheat which is rich in the minerals or the refined wheat flour? The authors should be succinct in their presentation of information

Line 89: Is the sourdough a probiotic or prebiotic? The authors have indicated that the production of extracellular polysaccharide by lactic acid bacteria makes sourdough a prebiotic. The authors should revise it. The probiotic potential could be due to the lactic acid bacteria. If so, do the authors think the baking temperatures will make the lactic acid bacteria survive after baking? If their reference is to the polysaccharide produced, at best that could rather be a “prebiotic” than a “probiotic”.

Line 96: Replace “paper” with “work”

Materials and Methods

I think it would have been great if the authors were able to indicate the level of polishing of the wheat grains prior to pulverising it into flours. The degree of polishing of course would impact on the levels of minerals in the flour. That could be a recommendation or something to consider for the future.

Table 1. The authors should specify the flour types used. That is whether they were “refined flours” or “wholegrain flours”. Also, Tables must always stand-alone. Consequently, the authors should indicate the names of the “yeast, sourdough and mixed leavening” at the bottom of the Table.

Line 132-134: The authors should correct the “H2O” to “H2O”. 

Results and Discussion

Table 2. The authors should indicate if the alphabets indicated in superscript in a row are significantly different. The authors should indicate if the values indicated are “Means” and their “Standard deviation”. These can be indicated at the bottom of the flour abbreviations in the Table. For example, under FWM, the authors can indicate Means (SD). The SD can be defined under the Table.

Generally, the introduction of the Tables and Figures in the manuscript should be at where they are first mentioned.

Figures must stand-alone. Consequently, the authors should expand the abbreviations used to represent the various flour types.

Line 238-239: The authors should revise this. The sentence should not stand in isolation.

Line 317: The authors did not present any Table showing the correlation coefficient.

Line 337: Could the authors provide the values of the minerals that were compared to those reported for this study?

Line 356-358: The authors should kindly explain the potential rationale behind this observation “The average values for the P, Mg, Ca, Fe, Mn, Zn and Cu observed in these 16 pre-baked breads were slightly lower than those obtained in the refined breads (BM and BC).”

Author Response

Mineral content in different wheat flours and bread varieties

Reviewer 1:

I am grateful for the opportunity to review this piece of work. In this work, Fernández-Canto et al. determined the mineral composition of ancient wheat cultivars from Castilla variety (refined, ground in industrial mills) and the Caaveiro variety (ground in stone mills) used either alone and mixed together and their corresponding bread formulated from these wheat flours. The findings of the study are important as hidden hunger (micronutrient deficiencies) is common public health challenge. Bread is a commonly consumed staple food across the globe. Consequently, it could be targeted as a promising medium for the delivery of micronutrients to address nutritional challenges associated with micronutrients despite its rich caloric contribution from its carbohydrate provision. Please, find below my comments for your perusal.

First of all, we would like to thank the reviewer for all their comments to improve the work submitted to “Foods” and titled “Mineral content in different wheat flours and bread varieties”. Based on your comments, we have made some modifications to the original paper. The changes made are shown below

Abstract

The wheat flour and bread formulations should be expanded and highlighted such as presented in the method “(a) 100% Castilla variety (FC), b) 100% Caaveiro variety (FCv), c) mixed flour (FM) (75% Castilla + 25% Caaveiro)”. The method for the mineral analysis should also be stated. For the results, the authors should state the concentration of predominant minerals that were identified in the flour from the wheat varieties and their corresponding bread.

We have changed the abstract based on your comments.

Introduction

Line 29: The authors should introduce “of” before “the”

We have introduced “of” before “the” (now line 33).

Line 43: The authors should revise this “As for micronutrients, cereals include mainly B vitamins,…………….”. It is wholegrain cereals that are rich in micronutrients, antioxidants and phytosterols from the bran oil.

            This point is clarified (now line 47).

Line 45: Revise “Furthermore, the wheat grains deriving from ancient wheat varieties….” to “Furthermore, wheat grains derived from ancient wheat varieties…………”

It was modified (now line 49).

Line 45-46: Please, provide a reference for this statement “Furthermore, the wheat grains deriving from ancient wheat varieties are richer in resistant starch, fiber, minerals, and phytochemicals than modern varieties.”

          The reference for this statement is the number 5, included at the end of the paragraph

Line 76-77: Is it the whole wheat which is rich in the minerals or the refined wheat flour? The authors should be succinct in their presentation of information.

The paragraph “In contrast, the greatest amount of phytates is found in bran. Thus, the breads made with wholemeal flour not only have a higher content of minerals, but also phytates, and the bioavailability of some minerals is reduced [10]. Other studies suggest that the increase in the concentration of minerals outweighs the negative effect induced by phytates on bioavailability [11]” was modified for clear beginning (now lines 67-71).

Line 89: Is the sourdough a probiotic or prebiotic? The authors have indicated that the production of extracellular polysaccharide by lactic acid bacteria makes sourdough a prebiotic. The authors should revise it. The probiotic potential could be due to the lactic acid bacteria. If so, do the authors think the baking temperatures will make the lactic acid bacteria survive after baking? If their reference is to the polysaccharide produced, at best that could rather be a “prebiotic” than a “probiotic”.

  You're right. It's a mistake. We have changed probiotic to prebiotic, which is the correct term (now line 93).

Line 96: Replace “paper” with “work”

It was modified (now line 100).

Materials and Methods

I think it would have been great if the authors were able to indicate the level of polishing of the wheat grains prior to pulverising it into flours. The degree of polishing of course would impact on the levels of minerals in the flour. That could be a recommendation or something to consider for the future.

Thank you very much for your comment, to take it into account in future research.

Table 1. The authors should specify the flour types used. That is whether they were “refined flours” or “wholegrain flours”. Also, Tables must always stand-alone. Consequently, the authors should indicate the names of the “yeast, sourdough and mixed leavening” at the bottom of the Table.

We have specified the types of flour used. That if they were “refined flours” or “wholemeal flours. We have included the explanation of the names of the breads at the bottom of the Table for greater clarity of the table

Line 132-134: The authors should correct the “H2O” to “H2O”.

We have modified “H2O” to “H2O”. and also, HNO3 to HNO3 (now lines 139, 140,142).

Results and Discussion

Table 2. The authors should indicate if the alphabets indicated in superscript in a row are significantly different. The authors should indicate if the values indicated are “Means” and their “Standard deviation”. These can be indicated at the bottom of the flour abbreviations in the Table. For example, under FWM, the authors can indicate Means (SD). The SD can be defined under the Table.

  The title of the table we indicated that: “Different letters indicate statistically significant differences according to a post-hoc Yuen's test”. We have put it at the bottom of the Table. We hope that this is clearer. We have indicated at the bottom of the flour abbreviations in the Table mean and standard error.

Generally, the introduction of the Tables and Figures in the manuscript should be at where they are first mentioned.

We have checked that the introduction of the Tables and Figures in the manuscript are where they are mentioned for the first time.

Figures must stand-alone. Consequently, the authors should expand the abbreviations used to represent the various flour types.

We have included the figures in the manuscript, following the instructions of the journal.

In all ways we have enlarged the figures.

Line 238-239: The authors should revise this. The sentence should not stand in isolation.

We have reviewed this. The sentence is not now isolated (now lines 232-235).

Line 317: The authors did not present any Table showing the correlation coefficient.

            Only one correlation coefficient was calculated, its value appears in the paragraph, 0.89. Samples in the dendrogram (tree) are linked together by hierarchical clustering based on their similarity. The Euclidean distance of mineral scaled data was used for this cluster policy. If you want to assess that the distances (i.e., heights) in the tree reflect the original distances accurately, you might compute the correlation between the cophenetic distances (heights) and the original distance data. As is the case of this paper, when the value of the correlation coefficient is closer to 1, the clustering solution reflects the data accurately.

Kassambara, A. (2017): Practical Guide To Cluster Analysis in R. Unsupervised Machine Learning. Published by STHDA (http://www.sthda.com), Alboukadel Kassambara.

Line 337: Could the authors provide the values of the minerals that were compared to those reported for this study?

The numerical values of the mentioned study have not been included since in this study the authors analyzed breads with different types of fermentation, among which one was made with spontaneous fermentation and another with commercial yeast. We believe that there would be too much data together in the text that could lead to confusion. Furthermore, the data reported in that study were expressed on a dry basis and our value tables are expressed on a wet basis. To make that claim, we compare them "internally" to our results on a dry basis.

In fact, when we review the bibliography, we observe that there is no clear criterion to express the results of minerals in flour and bread, since in some cases they are expressed on a dry basis and in others on a wet basis, and the moisture values are not always included. This fact makes it difficult to compare and discuss the results.

Line 356-358: The authors should kindly explain the potential rationale behind this observation “The average values for the P, Mg, Ca, Fe, Mn, Zn and Cu observed in these 16 pre-baked breads were slightly lower than those obtained in the refined breads (BM and BC).”

In the aforementioned study, 16 breads were made with different extraction rate (all of them refined), different leavening agents, different leavening and baking times, although none of them exactly coincided with the conditions in our study, therefore we decided to make an average of those 16 loaves to see if our results (refined breads) were below, above, or similar to their mean values. In this case, the data were also expressed on a dry basis.

Other changes: We have replaced Caaveiro throughout the text with 'Caaveiro'; native variety by autochthonous cultivar; ecotype by cultivar; and native flour by local flour. We believe that this way the manuscript is clearer.

Reviewer 2 Report

The manuscript investigated macro- and microelements in wheat flours and corresponding breads. It is of great interest to gain knowledge on the mineral content of traditional as well as modern wheat varieties. However, some questions remain after reading the manuscript:

Does the bread mineral content correlate with the flour? Why was it necessary to investigate also bread and not only flour (or vice versa). What changes in mineral content do you expect from varying breadmaking procedure (fermentation time, leavening agent)? This point was not discussed a single time and it seems a needless effort. (Variations in mineral content between breads made of one flour variety are most likely caused by moisture content (different baking loss)).

Introduction

L 41: What does “energy intake of 50-80%” refer to?

L 41-43: Change % (protein, lipid content) to g/100 g.

L 68: “P” is not a mineral, but an element

L 74: Change “minerals” to elements

L 92-99: Reasons not clear why fermentation conditions and leavening agent were varied in breadmaking. And why breadmaking is necessary at all.

Materials and Methods

L 102-107: Specify origin of flours.

L 104: What material is the stone made of? Can the mineral content of the flour be affected by stone abrasion/attrition?

L 107: Please explain “homologeous mixed wholegrain flour”. Is this a mixture of 75% Castilla + 25% Caaveiro, but other milling conditions (wholegrain)? Provide specifications.

L 108-110: Was the sourdough self-made or purchased? If applicable, explain processing steps or give information on supplier.

L 112: Salt = NaCl? add supplier and specification

L 117: Was the flour conditioned to a specific moisture? Because: When the flours differ in moisture, the results (elements) on fresh weight are not comparable. Corresponding to this: How was the amount of water required in the formulations determined?

L 121: What are “bread doughs”?

L 123: 30 min kneading will destroy gluten structure. Please specify kneading program. Are resting periods included?

L 124: “Bifinett” add manufacturer

L 125: Specify humidity

L 125/125: Add dough mass per loaf. “formed” means manually round shaped? Define cooling/ handling after baking.

L 129/130: Specify AOAC method! Add drying temperature. How was the bread treated prior drying? Completely milled with crust and crumb?

L 134: macro-

L 135: replace minerals with elements

Results

Tab. 2: “Different letters” in a row indicate…; Consider to recalculate elements on a defined moisture content (e.g. 10% for all flours)

discuss: FM = 0,25 * FCv + 0,75 * FC, the analysed values for FM are expected and can be calculated (and except for Cu it fits very well) à there is no need to discuss FM results that detailed for every element

Fig 1: Increase font size (especially loadings)

L 230: Give examples of “great diversity”

L 236/238: Substantiate “lower values” with specific values

L 238: Error in sentence

L 253-255; 261/263; 269; 277; 357-363; throughout the manuscript: Give values to the literature cited. Not only “smaller”, “lower” “in the range”

generally: Check throughout the manuscript, where “value” can be deleted. E.g. L 258/259: Mg and Zn were found …

L 266: Vignola et al. [30] reported…

L 264: Regarding Ca (Table 2), the highest and lowest contents…

L 272: Error in sentence

L 273: What are “large differences” concerning Fe?

L 276: check language “in a previous researcher”

L 280: an instead of “in”

Fig. 2: Explain abbreviations/sample code (readers must understand each Figure/Table without the manuscript text).

Tab. 3: Reorder elements corresponding to Tab. 2. Than it is easier to capture what’s not significant

Fig. 3: Provide better resolution! Increase font size of scores and loadings

L 366: PC1 77.1% but Fig. 3 displays 67.2%

L 373: So please explain the necessity to bake breads when the results are expected and similar to flour analyses.

Conclusions

Conclusion section is rather a summary than a conclusion. Suggestion: What about bioavailability of the minerals?

Author contributions: Check Vancouver criteria for authorship. M.N. Fernández-Canto and S. Boado-Crego are not eligible only because of doing investigations.

Author Response

 Reviewer 2
First of all, we would like to thank the reviewer for all their comments to improve the work
submitted to “Foods” and titled “Mineral content in different wheat flours and bread
varieties”. Based on your comments, we have made some modifications to the original paper.
The manuscript investigated macro- and microelements in wheat flours and corresponding
breads. It is of great interest to gain knowledge on the mineral content of traditional as well
as modern wheat varieties. However, some questions remain after reading the manuscript:
Does the bread mineral content correlate with the flour? Why was it necessary to investigate
also bread and not only flour (or vice versa). What changes in mineral content do you expect
from varying breadmaking procedure (fermentation time, leavening agent)? This point was
not discussed a single time and it seems a needless effort. (Variations in mineral content
between breads made of one flour variety are most likely caused by moisture content
(different baking loss)).
It refers to Regarding macronutrients, cereals stand out for their energy intake as
carbohydrates
Introduction
L 41: What does “energy intake of 50-80%” refer to?
It refers to 50-80% of the energy intake are carbohydrates. It was rewritten (now lines
44-45).
L 41-43: Change % (protein, lipid content) to g/100 g.
It was modified (now lines 45 and 47).
L 68: “P” is not a mineral, but an element.
It was modified.
L 74: Change “minerals” to elements
It was modified.
L 92-99: Reasons not clear why fermentation conditions and leavening agent were varied in
breadmaking. And why breadmaking is necessary at all.

2
Breadmaking allows cereals to be transformed into more palatable products for humans.
However, the operating conditions, both in terms of fermentation times and leavening
agent, vary greatly between manufacturers, giving rise to different products from a
nutritional and sensory point of view. Due to this, different conditions were tested to
see the influence on the element content.
Materials and Methods
L 102-107: Specify origin of flours.
The origin was specified (now lines 108-109).
L 104: What material is the stone made of? Can the mineral content of the flour be affected
by stone abrasion/attrition?
The stone mills are made of granite, very frequent in Galicia-NW Spain. We consider
that there is no transfer of minerals since the grain-stone contact time is fast.
L 107: Please explain “homologeous mixed wholegrain flour”. Is this a mixture of 75%
Castilla + 25% Caaveiro, but other milling conditions (wholegrain)? Provide specifications.
Homologeous mixed wholegrain flour is a mixture of 75% Castilla wholegrain + 25%
Caaveiro wholegrain. It was explained in the manuscript (now line 113).
L 108-110: Was the sourdough self-made or purchased? If applicable, explain processing
steps or give information on supplier.
We made it by mixing the corresponding flour, hydrated to 75% and leaving it at 25ºC
for 3 days. From the 3
rd day it was renewed every day, taking half the weight and
completing the same weight with new water and flour. It was waited for it to double in
volume and kept refrigerated until the following day. Explanation included in the text.
(now lines 129-132).
L 112: Salt = NaCl? add supplier and specification
Commercial NaCl, since bakers use commercial salt (now line 126).
L 117: Was the flour conditioned to a specific moisture? Because: When the flours differ in
moisture, the results (elements) on fresh weight are not comparable. Corresponding to this:
How was the amount of water required in the formulations determined?

3
In the manuscript the moisture content of each sample has been indicated (table 2),
from our point of view this avoids possible confusion.
In all recipes the flour was hydrated at 75%.
L 121: What are “bread doughs”?
This sentence was deleted, the correct sentence is: “The resulting breads are shown in
Table 1” (now lines138).
L 123: 30 min kneading will destroy gluten structure. Please specify kneading program. Are
resting periods included?
Kneading makes the gluten network more "elastic" improving the texture of the final
product. The breadmaking machine was only used to perform the kneading. NO
specific program has been used, since they do not coincide with the production
guidelines included in the PGI Pan Galego. Therefore, after kneading in the mixer, the
dough was left to rest in a container for the established times, the bread was shaped
(complete) and, finally, it was baked. The fermentation and baking conditions (time
and temperature) are specified in the manuscript in lines 133-137.
L 124: “Bifinett” add manufacturer
The manufacturer is Bifinett.
L 125: Specify humidity
The humidity value was included (line 135).
L 125/125: Add dough mass per loaf. “formed” means manually round shaped? Define
cooling/ handling after baking.
For each type of bread, the total amount of dough was 500 g. This clarification was
included in the lines 127-128. Loaves were manually round shaped. This fact was
clarified in lines 136.
L 129/130: Specify AOAC method! Add drying temperature. How was the bread treated prior
drying? Completely milled with crust and crumb?
We dried crust and crumb.
The AOAC method and temperature were specified (now lines 145-146).

4
L 134: macroIt was replaced (now line 150).
L 135: replace minerals with elements
It was replaced here and throughout the text.
Results
Tab. 2: “Different letters” in a row indicate…; Consider to recalculate elements on a defined
moisture content (e.g. 10% for all flours)
As we mentioned before, by including the humidity of each sample, possible confusion
is avoided and thus these results can be compared with other results in a fresh sample
or in a dry sample.
discuss: FM = 0,25 * FCv + 0,75 * FC, the analysed values for FM are expected and can
be calculated (and except for Cu it fits very well)
there is no need to discuss FM results
that detailed for every element
Although it is true that this analysis would not be strictly necessary, it is a way to check
that the results of the analyzes are correct. This mixture stands out because it is the
minimum required for the PGI Pan Galego.
Fig 1: Increase font size (especially loadings)
Due to the proportions of the graphic, the program code does not allow increasing the
font size.
L 230: Give examples of “great diversity”
With “great diversity” we mean that the results for the same element vary from one
investigation to another and that they may be due to different edaphological, climatic
conditions, etc. When we make the discussion by elements, the disparity in the results
can be observed.
L 236/238: Substantiate “lower values” with specific values
We did not put numeric values to lighten the text. If any reader wishes to see the values,
they can consult them in the reference. In addition, as we have already commented to
other referees in the bibliography, sometimes the results are given in a dry sample and
others in a fresh sample because "internal calculations" were necessary to make the

5
comparison and discussion, and we believe that putting numerical values would lead
to confusion.
L 238: Error in sentence
The sentence was rewritten (now lines 243-244).
L 253-255; 261/263; 269; 277; 357-363; throughout the manuscript: Give values to the
literature cited. Not only “smaller”, “lower” “in the range”
Above we already indicated the reasons for not putting numerical values
Generally: Check throughout the manuscript, where “value” can be deleted. E.g. L 258/259:
Mg and Zn were found …
We removed “value” where possible.
L 266: Vignola et al. [30] reported…
It was modified.
L 264: Regarding Ca (Table 2), the highest and lowest contents…
“Content” was deleted.
L 272: Error in sentence
The sentence was rewritten (now lines 276-278).
L 273: What are “large differences” concerning Fe?
As discussed earlier, we try to avoid numeric values for the reasons already stated. We
consider large differences when at least twice the value with which we compare.
L 276: check language “in a previous researcher”
It was revised
L 280: an instead of “in”
It was modified (now line 285).
Fig. 2: Explain abbreviations/sample code (readers must understand each Figure/Table
without the manuscript text).
It was included in the new version.
Tab. 3: Reorder elements corresponding to Tab. 2. Than it is easier to capture what’s not
significant
They are in a different order because in table 3 they were ordered based on the p-values.
6
Fig. 3: Provide better resolution! Increase font size of scores and loadings
Like the figure 1 due to the proportions of the graphic, the program code does not
allow increasing the font size.
L 366: PC1 77.1% but Fig. 3 displays 67.2%-
There is certainly an error in the text, the correct value is the one on the graph. It is
modified (now line390.)
L 373: So please explain the necessity to bake breads when the results are expected and
similar to flour analyses.
The behavior is expected to continue the same trend (and this is confirmed by the
results) although it is intended to analyze whether the breeadmaking process somehow
affects the mineral content.
Conclusions
Conclusion section is rather a summary than a conclusion. Suggestion: What about
bioavailability of the minerals?
For us the main conclusion is: ‘Caaveiro’ flour variety, an autochthonous Galician
cultivar has a nutritional quality near to wholemeal flour regarding elements. We
consider that it is important to deepen the knowledge of the nutritional characteristics
of local varieties to value them.
Regarding bioavailability, we have planned to conduct in vivo studies to clarify this
issue, especially in terms of traditional cereal varieties.
Author contributions: Check Vancouver criteria for authorship. M.N. Fernández-Canto and
S. Boado-Crego are not eligible only because of doing investigations.
These authors participated in the performance of the experimental work and gathered
the data, of course we consider that they are authors of the work.

Reviewer 3 Report

The article entitled “Mineral content in different wheat flours and bread varieties” is original.

Establish whether the mineral content of flour is affected by flour variety or use wholemeal flour, both in flour as a raw material, and in a manufactured product such as bread is of interest to both the healthy population and also patients with certain pathologies.

Minor revision should be made:

Line 34:  “Wheat is the cereal that is used the most in breadmaking”. I suggest “Wheat is the cereal for excellence in breadmaking”

On Section “2.2. Breadmaking procedure” Do not specify how many breads were made with each flour.

Table 2. Its necessary to specify what FWM, FCv, FM, FC means. Tables must be read by themselves

On Statistical analysis section I should resume all used methods. 

Author Response

Mineral content in different wheat flours and bread varieties

Reviewer 3:

The article entitled “Mineral content in different wheat flours and bread varieties” is original.

Establish whether the mineral content of flour is affected by flour variety or use wholemeal flour, both in flour as a raw material, and in a manufactured product such as bread is of interest to both the healthy population and also patients with certain pathologies.

First of all, we would like to thank the reviewer for all their comments to improve the work submitted to “Foods” and titled “Mineral content in different wheat flours and bread varieties”. Based on your comments, we have made some modifications to the original paper. The changes made are shown below

Minor revision should be made:

Line 34:“Wheat is the cereal that is used the most in breadmaking”. I suggest “Wheat is the cereal for excellence in breadmaking”

It was modified (now lines 33-34).

On Section “2.2. Breadmaking procedure” Do not specify how many breads were made with each flour.

       One loaf of each type of bread has been made. This data has been included in the manuscript (now line 121).

-Table 2. Its necessary to specify what FWM, FCv, FM, FC means. Tables must be read by themselves.

            We have completed the table 2.

-On Statistical analysis section I should resume all used methods.

Although we are aware that the statistical analysis process is a bit extensive, we believe that a summary of the statistical methods used would not allow us to see the purpose of each of them. We really believe that it is essential to explain the different treatments so that it is perfectly clear and that the readers of the article can understand it and, therefore, contribute to the knowledge of the statistical techniques used and that can be replicate. Even so, we have reduced this section.

Other changes: We have replaced Caaveiro throughout the text with 'Caaveiro'; native variety by autochthonous cultivar; ecotype by cultivar; and native flour by local flour. We believe that this way the manuscript is clearer.

Round 2

Reviewer 2 Report

Dear Authors,

thank you for considering the suggestions. From my point of view there are still some points that need to be addressed.

-        The aim of the study design is still not clear. Please answer the questions (also in the manuscript). As it seems now, it is completely redundant to analyse breads for mineral content only. The minerals do not disappear during processing. If you only want to show that the breads are rich in minerals, one formulation is totally sufficient. So why the effort of varying fermentation? Please explain the ideas behind the design.

Why was it necessary to investigate bread and not only flour (or vice versa). What changes in mineral content do you expect from varying breadmaking procedure (fermentation time, leavening agent)?

-      Information on suppliers of flour and salt are still not included

-       If it is not possible to change figures in the original program, extract the data and create a figure manually in another program. It’s worth the effort.

Due to the proportions of the graphic, the program code does not allow increasing the font size

 -        Independent from each other, also Reviewer 1 suggested to include specific values from references. Please take this suggestion serious, as your answer seems to be an excuse to save time. Scientific readers are in the position to differentiate between dry matter and fresh weight. And if you would recalculate your results and present them as dry matter, it would simplify comparison.

We did not put numeric values to lighten the text. If any reader wishes to see the values, they can consult them in the reference. In addition, as we have already commented to other referees in the bibliography, sometimes the results are given in a dry sample and others in a fresh sample because "internal calculations" were necessary to make the comparison and discussion, and we believe that putting numerical values would lead to confusion

-          L 23: correct unit “P (441.0 g/100 g)”
